# Physical and Chemical Characterisation of the Pigments of a 17th-Century Mural Painting in the Spanish Caribbean

**DOI:** 10.3390/ma14226866

**Published:** 2021-11-14

**Authors:** Virginia Flores-Sasso, Gloria Pérez, Letzai Ruiz-Valero, Sagrario Martínez-Ramírez, Ana Guerrero, Esteban Prieto-Vicioso

**Affiliations:** 1School of Architecture and Design, Faculty of Social Sciences, Humanities and Art, Pontificia Universidad Católica Madre y Maestra (PUCMM), 2748 Santo Domingi, Dominican Republic; 2Instituto de Ciencias de la Construcción Eduardo Torroja (IETcc-CSIC), 28033 Madrid, Spain; gperezaq@ietcc.csic.es (G.P.); aguerrero@ietcc.csic.es (A.G.); 3School of Civil and Environmental Engineering, Faculty of Sciences and Engineering, Pontificia Universidad Católica Madre y Maestra (PUCMM), 2748 Santo Domingo, Dominican Republic; letzairuiz@pucmm.edu.do; 4Instituto Estructura de la Materia (IEM-CSIC), 28006 Madrid, Spain; 5Departamento de Inestigación, Universidad Nacional Pedro Henríquez Ureña (UNPHU), 1423 Santo Domingo, Dominican Republic; eprieto@unphu.edu.do

**Keywords:** Caribbean, Cathedral of Santo Domingo, 17th century, UV–VIS–NIR, Raman spectroscopy, XRD, SEM/EDX, pigment, mural painting, dating

## Abstract

The arrival of Spaniards in the Caribbean islands introduced to the region the practice of applying pigments onto buildings. The pigments that remain on these buildings may provide data on their historical evolution and essential information for tackling restoration tasks. In this study, a 17th-century mural painting located in the Cathedral of Santo Domingo on the Hispaniola island of the Caribbean is characterised via UV–VIS–NIR, Raman and FTIR spectroscopy, XRD and SEM/EDX. The pigments are found in the older Chapel of Our Lady of Candelaria, currently Chapel of Our Lady of Mercy. The chapel was built in the 17th century by black slave brotherhood and extended by Spaniards. During a recent restoration process of the chapel, remains of mural painting appeared, which were covered by several layers of lime. Five colours were identified: ochre, green, red, blue and white. Moreover, it was determined that this mural painting was made before the end of the 18th century, because many of the materials used were no longer used after the industrialisation of painting. However, since both rutile and anatase appear as a white pigment, a restoration may have been carried out in the 20th century, and it has been painted white.

## 1. Introduction

Pigments have been used in all cultures and civilisations of the world from the prehistoric times to the present. Pigments have always played a role in the human evolutionary process and are a form of expression over time. However, considering for example plasters, degradation process due to lack of maintenance, abandonment and oblivion have caused much of them to disappear, leaving only traces. When pigments with different chemical composition are applied to an object or a building, they interact with the binder, substrate, environment, etc., and as a whole evolve to become a part of it, its history and its construction system. With the recovery of dyes, information can be obtained about the period and techniques used to apply the pigments. There are different factors that can help to characterise the period from which they come, such as whether the pigments are local or imported, price, fashion, etc.

After comparing pigments with existing graphic and documentary information, their study can provide complementary information regarding architecture, culture and society. In the field of historical heritage, the characterisation of pigments allows the study of the historical evolution of a building and relates it to technical, constructive, aesthetic and social aspects. In addition, the study of dyes provides information on the maintenance and preservation of the building. Thus, if a plaster on a wall has several layers of various pigments, it is possible to have an idea of the number of interventions to which the wall has been submitted. For these reasons, pigments characterisation has become increasingly important for the restoration and maintenance of historical buildings [1,2].

In this way, one of the records referring to polychrome on stone observed on marble (statues of the Parthenon of Athens), was made by Edward Daniel Clarke (mineralogist) and Edward Dodwell (archaeologist). In addition, Gell and Gandy, also described the engraved and painted friezes of the Parthenon [3]. The aesthetic criteria and availability of materials, as well as the function of each artefact in the social and cultural environment determined the use and predilection of certain colours in the polychrome and painting of ancient Greece [4].

Colours are related to the capacities for reflection, absorption and transmission of visible light, which depend mainly on the chemical composition of the pigments used, although effects due to physical characteristics such as particle size are also expected [5]. The technological progress of the last few decades has favoured the study of colours and guaranteed the accuracy of results under certain measuring conditions. Although colour information provides an approach to the identification of pigments, colour detection may be influenced by the effects of environment on outdoor measurements, by the morphology in the case of objects with concavities and convexities, which complicate the analysis, or by material changes [6]. For example, illumination of an object from different orientations can give rise to different colour perceptions. On the other hand, chemical transformations may occur in wall paintings, as blue azurite being transformed into green malachite. This change is due to the presence of water, which may have seeped through walls or condensed on mural surfaces. Therefore, it is important to conduct complete physical, chemical and mineralogical analyses to determine the real pigments present in any studied object.

In order to select the colours that should be used in the case of buildings restoration in cities, several techniques were used for the analysis of façades polychrome to obtain a colour palette that defines the colour of each city and provides signs of its identity. These techniques included visible spectroscopy, to analyse the colours, and Fourier-transform infrared spectroscopy (FTIR) and energy-dispersive X-ray spectroscopy (EDX) coupled with scanning electron microscopy (SEM), to analyse pigment compositions [7,8]. In 2007, several gothic mural paintings (13–15th centuries) in the churches of Slovenia were analysed using samples obtained in stratigraphic sections, which were examined via digital photography, optical and electronic microscopies, IR spectroscopy and EDX for chemical analysis. Additional information concerning the pigments and mortars was obtained via X-ray diffraction (XRD) and IR transmission spectroscopy using KBr pellets [9]. 

The pigments in the colour layers on the walls of the Cistercian Abbey of Stična and the Manor of Novo Celje in Slovenia were also studied. Mural paintings were inspected via the combined application of different microscopic techniques, optical microscopy to determine their colour and scanning electronic microscopy for the structural features. EDX was used to determine elemental distributions on selected surfaces and elemental compositions of individual pigments [10].

In addition, the pigments, dyes, binders and techniques of Mayan mural paintings were investigated via atomic force microscopy (AFM), SEM combined with EDX, XRD, solid-state electrochemistry, visible spectrophotometry, ultraviolet–visible (UV-VIS) spectroscopy and gas chromatography/mass spectrometry (GC/MS), among others [11]. Similar techniques were also used for the analysis of colours and sun exposure in the Towers of Quart in Valencia, Spain [12,13].

In Spain, a chromatic study of the exterior facades of the Monastery of Our Lady of the Angels in Valencia was conducted to establish the most appropriate chromatic guidelines for its restoration. The investigation included a visual analysis of the remaining pigmentation, measurement of their chromatic parameters using specific instruments, statistical evaluation of the gathered information and physical formalisation of the chromatic charts in standard notation. The average colours obtained in the study, both on the Munsell scale and in the CIELab space, could be considered acceptable for restoration [14]. In Italy, facades were studied using spectrophotometry, spectroradiometry of small surfaces (from 1 to 10 mm^2^) and digital photography to verify the influence of variations in natural light on perceived colours [15]. To analyse samples of mural paint to determine the original formulation of a painting, instrumental techniques and traditional analytical chemistry, particularly FTIR, were used [16]. 

Raman spectroscopy has also been incorporated as a complementary technique for objective dyes and pigments assessment in the study of historical buildings. Several works published on the historical heritage of Italy have reported on efforts to recover the historical centre of Rome. Based on the limits and potentialities of a ‘colour plan’, morphological observations via optical microscopy in transmitted light and via Raman spectroscopy were conducted [17]. In addition, the chromatic alteration of Roman heritage in Aosta was studied using reflected visible light under an optical microscope and using micro-Raman measurements [18], whereas the painted silk panels of Palazzo Barberini in Rome were investigated through a combination of more traditional techniques with micro-Raman spectrometry (MRS) [19]. MRS was combined with FTIR microspectroscopy to identify the pigments of wall paintings from the Roman town of Emona in Slovenia [20]. On the other hand, in situ analysis using portable equipment was performed on wall paintings from the Islamic epoch (10th to 12th centuries) and Christian monarchy (14th to 16th centuries) at the Seville Alcazar in Spain [21].

In Latin America and the Caribbean several studies were carried out. In Mexico, pigments and other components of nine mural paintings were studied in three colonial Augustinian ex-convents. X-ray fluorescence (XRF) and Raman spectroscopy were used. The authors identified different pigments used for different dyes typical of Mexican Colonial colour palette [22]. While in Bolivia, the authors studied two samples extracted from the image of Our Lady of Copacabana to identify the corresponding pigments. MRS complemented with SEM-EDX, and high-performance liquid chromatography (HPLC) were used [23]. In the church of San Andrés de Pachama (Chile), micro samples were taken from one of the mural paintings and have been analysed using MRS complemented with SEM-EDX, attenuated total reflection infrared spectroscopy, XRD and HPLC [24].

In Dominican Republic, the colours of the Coralina limestone in the main facade of the Cathedral of Santo Domingo were studied by the authors using UV–VIS–NIR spectroscopy and Munsell systems [25]. While, in Colonial missions in Mexico the authors studied painted decorations located on the wood ceilings and walls. False colour infrared imaging, XRF and fibre-optics reflectance spectroscopy were used to characterise the elemental and molecular composition of the decorations [26]. Following the study of the Cathedral of Santo Domingo in Dominican Republic, this paper presents an analysis of a mural painting in the chapel of Our Lady of Mercy, built in 1602–1613. Five pigments with different colours were identified in the stratigraphic samples: ochre, green, red, blue and white. Quantitative colorimetric characterisation and assessment of the chemical compositions of the pigments were performed through both in situ and laboratory tests. This is the first study of a 17th-century mural painting in the Caribbean region and provides new insights into the historical evolution of the cathedral. 

## 2. Our Lady of Mercy Chapel in the Cathedral of Santo Domingo

The Caribbean island of Hispaniola was a Spanish colony for more than two centuries. At the beginning of the 16th century, the first cathedral in the Americas was built in the city of Santo Domingo. The cathedral has 14 side chapels built by important families of the colony, and by brotherhoods. One of these chapels was built in 1602–1613 by the Brotherhood of Our Lady of Candelaria, which belonged to the Black slaves Biafras and Mandingoes and had been installed in the cathedral since 1586. In fact, this was the first chapel constructed in the Cathedral of Santo Domingo that belonged to Black slaves [27]. 

At the beginning, this large and important brotherhood was composed only of Black slaves. However, when the trade of slaves, especially those of these ethnic groups, decreased, the brotherhood began to include Spanish brothers, mostly of Canary origin, because Our Lady of Candelaria is the patron saint of the Canary Islands. The brotherhood disappeared in the middle of the 19th century, after which the chapel was governed by the church that installed Our Lady of the Sacred Heart [28]. 

The chapel was built in two stages. In the first stage, which was at the beginning of the 17th century, the chapel was much smaller than its current form and was approximately 2 m deep. In the second stage, circa the 18th century, the chapel was enlarged to approximately 7 m deep, because the number of brotherhoods increased with the arrival of the Spaniards [29]. At the beginning of 2019, restoration work was carried out in the Chapel prior to the installation of Our Lady of Mercy. At that time, remains of a mural painting appeared that correspond to the initial stage in which the chapel was managed by the brotherhood of the Candelaria. Samples were selected taking into account different coloration, position and stratigraphy (Figure 1).

## 3. Materials and Methods

The visual analysis of the wall of the chapel, allowed detecting areas with different pigments, including an area with several layers of pigments located on an area 1.00 m wide by 1.50 m high in the lower-right zone of the inner wall of the chapel access arch (Figure 1a). This area was selected for the analysis, along with the remains of a blue-green pigment, located inside the pointed arch that frames access to the chapel. This second area, in which the pigment is dispersed at a number of small points (Figure 1b), was chosen to compare with the blue pigment located at the inner wall mural painting (Figure 1c). In the stratigraphy, the pigments can be visually observed as ochre, red and blue coloured in order towards the white outer surface. 

Four samples were taken, one of each pigment, from the areas where they were easily detached, so it was not necessary to use any tool that would damage the wall. One piece of each sample was analysed with all the available techniques. In the case of the red sample, a superficial white layer is observed, which was also analysed.

The first characterisation of the remains of the mural painting was performed in situ using non-destructive techniques. Specifically, optical characterisation of colour and reflectance was performed via UV–VIS–NIR spectrophotometry using a StellarNet portable fibre-optic spectrometer. The equipment consists of a halogen lamp with a colour-equalising filter, a detector covering a nominal wavelength range between 200 and 1080 nm (model BLK-C-SR) and a reflectance probe, in which seven illumination fibres are installed around a central reading fibre. The probe was placed using a suitable accessory at a fixed distance from the measuring surface and at a 90° angle. This is the most suitable configuration, based on the expectation that different areas with small dimensions and appreciable surface roughness must be analysed. The reflectance was measured with respect to that of a Spectralon pattern, with a reflectance greater than 97% over the entire measurement range of the equipment.

It is important to remark that the remains of green colour were small and located within the decoration of the access arch of the chapel. Consequently, the in situ characterisation was only possible in a specific area of the arch, which allowed reliable placement of the reflectivity measurement accessory. In addition, the characterisation of the white layer presents difficulties because it is the outer layer of the remains of the studied mural painting and, therefore, includes numerous areas affected by dirt and texture irregularity. For this reason, only two spectra were obtained for this layer.

From the reflectance spectra in the visible range (380–780 nm), the three colour coordinates defined in the CIELAB 1976 space, corresponding to the CIE 1964 Standard Observer and D65 Illuminant, were calculated. In this model, the a* coordinate may have values between −90 and +90, with negative values for green, and positive values for red. The b* coordinate may also have values between −90 and +90, where negative is for blue, and positive is for yellow. Finally, the L* coordinate indicates lightness, with values of 0 for black, and 100 for white.

In this first in situ study, Raman spectroscopy was also used for the chemical characterisation of the mural painting. Specifically, a portable DeltaNu Raman Inspector was used for this procedure. This system is equipped with a 785-nm diode laser for excitation, which has a maximum output power of 120 mW at the source, and a thermoelectrically cooled charge-coupled device detector with a range of 200–2000 cm^−1^. The instrument uses NuSpec software, which allows control over all spectrometer functions.

Samples of the four identified pigments were transported to laboratories to complete the physical and chemical characterisation of the mural painting. The pigments selected were green (ChG), red (ChR), blue (ChB) and white (ChW) (Figure 1). A number of samples were obtained from detached fragments of the mural painting. The ochre pigment was directly applied to the stone and its extraction required the use of tools that could damage the stone, so it was only analysed in situ.

The optical characterisation of the samples was performed in a laboratory with the same StellarNet portable fibre-optic spectrometer that was used on-site. This analysis is expected to give rise to more accurate reflectance spectra with respect to the on-site one, as flat and clean areas of the samples were selected to ensure optimal measurement conditions. More specifically, four spectra were measured at different points of the largest green fragment (ChG) which had a non-homogeneous green colour representative of the predominant shades in the remains, lighter than in the area measured in situ. For the red sample (ChR), spectra were obtained at five different points within the flattest, darkest and uniform parts. Two different small fragments were measured for the blue sample (ChB), and three spectra were obtained at different points within each fragment. Finally, the white outer layer (ChW) was characterised more reliably in the laboratory than in situ, as the cleanest and flattest areas of the samples were measured at five different points.

Characterisation via Raman spectroscopy was performed using a Renishaw inVia spectrometer equipped with a Leica microscope. The spectra were recorded using a 532-nm Nd:YAG laser in the range 4000–100 cm^−1^. The measurements were obtained with a laser output power of 5 mW, acquisition time of 10 s and five accumulations. The frequencies were calibrated using silicon. The four samples in Figure 1d were characterised in terms of chemical and mineralogical composition via X-ray diffraction (XRD) using a Bruker D8 Advance diffractometer operating at 40 kV and 30 mA, using the CuKα line with a step of 0.05°/s. In this study, the three aforementioned techniques were performed on the samples without need for any treatment.

In order to determine the binder presence in the samples, FTIR was performed. The apparatus used was a Bruker Vertex 70V spectrophotometer and transmission was measured by the KBr pellet method, in the 4000–400 cm^−1^ range, with 100 scans at a resolution of 2 cm^−1^.

Finally, a piece of each sample was subjected to microstructural characterisation. The individual pieces were coated with a conductive C layer and analysed using a Hitachi S-4800 scanning electron microscope. The microscope was equipped with a BRUKER 5030 X-ray analyser for elemental chemical characterisation via energy-dispersive X-ray spectroscopy (SEM–EDX). 

## 4. Results

As a summary, Table 1 shows the chemical and mineralogical characteristics of the samples, showing the clear correlation between the different information obtained by different techniques.

### 4.1. In Situ Characterisation

The reflectance spectra measured in situ on the sample of mural paintings in the Our Lady of Mercy Chapel are related to the pigments ochre, red, green, blue and white. The two spectra measured in the green zone (ChG) have a low value of reflectance, below 10% in the visible range, and a very faint and wide maximum in wavelengths about 530 nm (Figure 2a light green). The average colour coordinates calculated from these two spectra have a*/b* values equal to −6.2/+2.1 (Table 2), which indicate a green colour (Figure 3). On the other hand, the L* coordinate has an average value of 34.8, which indicates a relatively dark shade. These results are consistent with visual observations in the measured area.

The greatest number of measurements was obtained for the ochre layer, located at the base of the arch, because of the impossibility of extracting a sample of this layer to be analysed in the laboratory without affecting the integrity of the sample. Despite the different levels of reflectance, most of the spectra (solid lines) present common characteristics, with a relatively low reflectance for low wavelengths and a relative minimum at 480 nm (Figure 2b light ochre). From this wavelength, significant increases in reflectance, by up to values between 52% and 91% at 600 nm, were observed. In some spectra (dashed lines), these increases are more abrupt, and the reflectance for longer wavelengths remains almost constant up to 1050 nm. 

However, in all the spectra depicted in solid lines (Figure 2b light ochre), a wide relative maximum in wavelength, about 790 nm, was observed, with reflectance values varying between 71% and 85%. This behaviour is characteristic of pigments based on Fe, namely yellow ochre (FeO(OH)·nH_2_O) and goethite (α-FeO(OH)) compounds. Absorption due to ligand field transitions of the Fe^3+^ ion causes decreases in reflectance by approximately 650 nm and 920 nm. The relative minimum observed at 480 nm is also characteristic of the charge transfer transitions of the Fe–O bond in these compounds [30,31]. Because of the significant differences in the spectra, the values of the colour coordinates obtained exhibit a high dispersion. 

All the measured points exhibit high values of L*, with an average value of 80.2 ± 6.9, and positions near the positive b*-axis in the first quadrant of the b*-versus-a* representation (Table 2). This variety in behaviour corresponds to the variety in shades that can be observed in the analysed ochre layer.

Similarly, the spectra obtained on the red layer (ChR) present very different levels of reflectance (Figure 2c light red), which are consistent with the different shades of this layer, and which result in a high dispersion in the values of the L* coordinate. However, all these curves show common and specific characteristics of pigments based on anhydrous Fe oxides. These may be hematite (α-Fe_2_O_3_) or red ochre, which is a natural product coloured by the presence of anhydrous iron oxide Fe_2_O_3_ produced by the loss of water from yellow ochre [31,32]. The increase in reflectance starts at 550 nm and shifts to longer wavelengths with respect to those of the spectra measured in the ochre layer. The maximum reflectance is more pronounced in this red layer, varying between 21% and 73%, and was observed at a wavelength of approximately 765 nm, which is associated with the red colour of the surface being analysed. In the b*-versus-a* representation (Figure 3), the results for this layer are consistently below the bisector of the first quadrant.

All reflectance spectra measured on the blue layer (Figure 2d light blue) (ChB) have a maximum and minimum in the visible range. The maximum is almost symmetrical and occurs at approximately 465 nm, with appreciable variations in width and reflectance (between 26% and 43%). The minimum is at approximately 610 nm, with a variation in reflectance values between 7% and 25%. This behaviour is characteristic of an ultramarine-blue pigment, which is composed mainly of the mineral lazurite [33]. For the blue-layer spectra, an average L* coordinate of 46.3 ± 8.0 was obtained (Table 2), whereas the points in the b*-versus-a* representation extend below the bisector of the fourth quadrant (Figure 3).

Finally, the L* coordinate of the white layer has a value close to 100 (more precisely, 90.6 ± 1.3) (Table 2), whereas the a* coordinate is practically zero, which corresponds to a white colour (Figure 3). However, the value of b*, equal to 13.4 ± 2.6, indicates a yellowish colour, which must be due to the dirt on the surface.

The in situ optical characterisation provided information on the variety of pigments and shades that were identified in the painting. The colours of the ochre and red layers suggest that the pigments used for their preparation are based on iron compounds: yellow ochre or goethite and red ochre or hematite, respectively [30,31,32]. For the blue layer, the spectra are similar in terms of the visible range to that of the ultramarine-blue pigment, whose generic composition is (Na,Ca)_8_(AlSiO_4_)_6_(SO_4_,S,Cl)_2_ [33].

However, the characterisation must be completed with mineralogical identification of the pigments used in each sample. For this purpose, in situ characterisation was performed using Raman spectroscopy in samples ChO and ChR (Figure 4 in situ) . In both spectra , the strongest signal was observed at 1085 cm^−1^, which, together with other lower intensity signals (712 and 280 cm^−1^), is associated with the vibrations of the CO_3_^2−^ groups in calcite [34]. The presence of this calcium-carbonate phase suggests the use of a lime base for the application of the paint layers. The band appearing at 1050 cm^−1^ in sample ChR (Figure 4d in situ) may be due to white lead or to nitrates. In the former case a sharp band is expected and in the latter a broad band is expected, as it appears in the spectrum of the ChR sample, indicating that the band may be due to a mixture of nitrates [35]. The Raman spectra show no signs of pigment in any of the paints probably due to the low spectral resolution of the portable equipment [36].

### 4.2. Characterisation of Samples in the Laboratory

#### 4.2.1. Optical Characterisation

The reflectance curves of the green sample (Figure 2a dark green) show a wide maximum with a reflectance between 25% and 30% at a wavelength in the order of 530 nm. This value of maximum reflectance in the visible range corresponds to the green colour that characterises the sample. Green pigments used throughout history originated from various minerals such as glauconite, celadonite, copper acetates (verdigris), copper basic carbonate (malachite), atacamite and related compounds [31]. These pigments may be distinguished from the shape of the UV–VIS reflectance spectra and the position of the minima related to absorption bands. In fact, the shape of the broad maximum and the low reflectance along the 700 to 900 nm wavelength range suggest that the green pigment corresponds to malachite [37]. 

Spectra of the red sample (ChR) show a nearly constant reflectance on the order of 12% for wavelengths below 550 nm (Figure 2c dark red). The reflectance increases to a well-defined maximum, with a value of 44–51%, at approximately 765 nm, followed by a minimum, with a reflectance of 39–45%, at approximately 880 nm. As revealed through the analysis of the in situ measurements, these spectra may correspond to iron-oxide-based pigments. These results are consistent with those for the hematite phase (α-Fe_2_O_3_) [31,32].

The reflectance curves of the blue sample (ChB) present characteristics already observed in the in situ characterisation, demonstrating a wide maximum, with values between 26 and 43% at 465 nm, within the range associated with its blue colour (450–495 nm) (Figure 2d dark blue). As indicated previously, the spectra are similar in terms of the visible range to that of the ultramarine-blue pigment, with a generic composition of (Na,Ca)_8_(AlSiO_4_)_6_(SO_4_,S,Cl)_2_ [33].

Finally, the white outer layer (ChW) was characterised more reliably in the laboratory than in situ. The cleanest and flattest areas of the samples were measured at five different points. The spectra obtained are practically flat and have a high value of reflectance, greater than 80% over the entire wavelength range (Figure 2e dark black). These spectra are consistent with the white colour of the sample. White pigments may be identified by the slope of the reflectance spectrum in the 350–450 nm wavelength range [38]. According to the derivative curve of the reflectance spectrum, no peaks occur in this range for the lead-white pigment, whereas maximum positions occur at approximately 385 and 408 nm for the zinc-white and titanium-white pigments, respectively. The first derivative of the spectra measured on the ChW sample shows a maximum at approximately 405 nm, indicating that the pigment in the white layer is based on titanium dioxide.

The colour coordinates in the CIELAB space were calculated from the reflectance spectra measured in the laboratory. The results are similar to those obtained from in situ measurements, although in the latter case, the dispersion of values, which represent the variety of shades observed in the samples of each colour, is significantly greater. The average colour coordinates obtained in the laboratory and outlined in Table 2 constitute a quantitative characterisation of the predominant pigments in the samples of mural paintings in the Our Lady of Mercy Chapel of the first cathedral of the Americas.

#### 4.2.2. Raman and FTIR Spectroscopy

The four samples were analysed in the laboratory via micro-Raman spectroscopy with a laser wavelength of 532 nm. For each sample, Raman spectra were obtained from the pigmented face and posterior face. Small pieces of the different pigments were selected (1–2 mm) and placed directly on the sample holder of the Raman spectrophotometer. For the micro-Raman spectrum of the green ChG sample (Figure 4a in laboratory), the strong signals observed in the ranges 4000–3000 cm^−1^ to 3439, 3353 and 3308 cm^−1^ were attributed to the vibrations of Cu_3_≡O–H---Cl. The signals observed at 929, 894, 839 and 802 cm^−1^ are due to the deformation vibrations of hydroxyl groups, whereas the strong signal at 514 cm^−1^ is due to Cu–O stresses. Finally, the strong signal at 399 cm^−1^ was attributed to the stretching vibrations of the Cu–Cl joints. Of the three basic copper (II) chloride minerals with the same stoichiometry Cu_2_Cl(OH)_3_, i.e., atacamite, paratacamite and clinoatacamite, considered to be pigments from antiquity [39], not a single one could be easily identified in the sample. On the other hand, a number of signs indicating azurite (3425; 1577; 1429; 929; 839; 764; 401; 332; 282; 265; 247; 179 and 153 cm^−1^) were identified. The presence of atacamite, paratacamite and clinoatacamite can be a result of the degradation of original paint pigments, such as malachite and azurite, in environments containing chlorides [40]. It should be noted that until 2008 the Cathedral did not have air conditioning and cooling was achieved by opening doors, which allowed the entry of outside air with pollution from nearby vehicles and industries. There are no pollution meters in Santo Domingo, but a study carried out in 2013 revealed that the amount of SO_2_ and NO_x_ was above the values allowed by the WHO [41].

The bands observed at 530; 430; 266 and 179 cm^−1^ can be attributed to the presence of malachite (Cu_2_CO_3_(OH)_2_), which may be added by the artist or come from the degradation of the azurite(Cu_3_(CO_3_)_2_(OH)_2_).

Analysis of the back of the ChG sample via Raman spectroscopy (not shown) revealed significant fluorescence [34]. Thus, it was difficult to identify the Raman signals, and only the strongest signal could be identified, through the vibrations of the CO_3_^2−^ groups present in the calcite phase of calcium carbonate (1084 cm^−1^). This result suggests that the paint was applied on a lime base.

Because of the high fluorescence of the micro-Raman spectrum of the red pigment (ChR), only the signals at frequencies below 600 cm^−1^ could be identified (Figure 4c, in laboratory). In this range, signals at 609, 435, 408, 289, 243 and 222 cm^−1^, which are associated with hematite iron oxide (Fe_2_O_3_), were identified [42]. On the other hand, the signals at 606 and 435 cm^−1^ frequencies may also be due to the presence of TiO_2_ as rutile [43]. Other areas of the sample included white deposits, whose micro-Raman spectrum shows a strong signal at 143 cm^−1^, indicating the presence of anatase TiO_2_. The addition of this white pigment is usually associated with mixing for a lighter dye. The Raman spectrum on the back of the ChR sample indicates that the base of the paint was probably lime [34], as in the previous samples.

The micro-Raman spectrum of the ChB sample (Figure 4e in laboratory) shows the characteristic vibration signals of lazurite, Na_3_Ca(Si_3_Al_3_)O_12_S [44,45,46], the main component of the mineral lapis lazuli. Its most intense Raman signal was observed at 546 cm^−1^ (symmetrical stress, *v*_1_ S^3^^−^), together with another of medium intensity, with maximum at 256 cm^−^^1^ (*v*_2_, strain vibration S^3^^−^). Another series of signals were observed; these signals were identified to be harmonics, characteristic of the resonant Raman effect (Figure 4e in laboratory) [47]. These bands appear at double frequency (1092 cm^−1^) and triple frequency (1648 cm^−1^) of the most intense frequency (546 cm^−1^), and at other frequencies (802 cm^−1^) by combinations of the signals at 256 and 546 cm^−1^. Analysis of the back of the sample via micro-Raman spectroscopy revealed, as in the two previous samples, that the pigment was probably applied on a lime base.

Finally, the micro-Raman spectrum of the white ChW sample (Figure 4f, in laboratory) shows two intense signals with maxima at 609 and 444 cm^−1^ due to vibrations of the Ti–O link of rutile TiO_2_ [40]. As was performed for the rest of the samples, the back of the ChW sample was analysed, and the spectrum, despite showing a high level of fluorescence, made it possible to identify the presence of CaCO_3_ [34]. 

FTIR spectra of the pigments (Figure 5) allow to identify the presence of calcite by the bands at 1449 cm^−1^ (*v*_3_), 875 cm^−1^ (*v*_2_) and 712 cm^−1^ (*v*_4_); the presence of gypsum by the bands corresponding to water at 3550, 3492, 3405 cm^−1^ (*v*_water_) and 1692, 1627 cm^−1^ (δ_water_) and to sulphate at 1111 cm^−1^ (*v*_3_) and 669, 600 cm^−1^ (*v*_4_), as well as the presence of anhydrite (1153 cm^−1^) in all of them. The spectrum shows, as well, bands in the range 3000–2800 cm^−1^ due to symmetric and asymmetric C-H stretching of CH_3_ and CH_2_ groups [48]. These signals are due to the presence of organic matter and are more intense in the ChR sample with a higher kaolinite content, so it can be associated with this mineral. A sharp signal of medium intensity with a maximum at 2513 cm^−1^ is also observed, which some authors associated to vibrations of CO_3_^2−^ in calcite and minerals of the calcite and dolomite groups [49]. 

Only some signals that could correspond to the pigments were identified in the case of sample ChR (signal at 536 cm^−1^ from hematite) and sample ChW (signal at 517 cm^−1^ from TiO_2_). In these same samples, kaolinite was also identified [50].

#### 4.2.3. X-ray Diffraction (XRD)

X-ray diffractograms were obtained for the four samples analysed in the laboratory (Figure 6). In all the cases, the presence of calcite crystalline phase (CaCO_3_) (C) can be identified, with a maximum intensity peak at 2θ 29.45°. In the case of the green sample (ChG), the diffractogram shows low-intensity signals. The most intense peaks are characteristic of the paratacamite phase, one of the four polymorphic structures of copper oxychloride (Cu_2_Cl(OH)_3_) that corresponds with the strong signals observed in Raman spectroscopy attributed to the vibrations of Cu_3_≡O–H---Cl. The red-sample diffractogram (ChR) confirms the presence of hematite (α-Fe_2_O_3_) which provides the red colour. Diffraction peaks characteristic of anatase (2θ 25.69°) and rutile (2θ 27.44°) polymorphs of titanium oxide (TiO_2_) could be detected in this diffractogram that it is in accordance with the micro-Raman spectrum that shows a strong signal at 143 cm^−1^, indicating the presence of anatase. This white phase could have been added to lighten the red colour of hematite. In addition, the calcite phase was the only phase observed in the diffractogram of the blue sample (ChB), suggesting that the pigment used in this case has an amorphous character. Finally, the diffractogram of the white sample (ChW) confirmed the presence of rutile TiO_2_, with two maximum intensity peaks at 2θ 27.44° and 54.32°, which was also inferred from its micro-Raman spectra at 609 cm^−1^ and 444 cm^−1^.

#### 4.2.4. Scanning Electron Microscopy and X-Ray Microanalysis (SEM–EDX)

The morphology, microstructure and composition of each sample were analysed via SEM–EDX. Figure 7 shows three images of each sample: the general morphology observed at 1500× magnification, magnified image with the different morphologies identified and details of the predominant microstructure with the characteristic EDX spectrum.

In the green sample (ChG), the predominant structure is formed by crystals in the form of flat plates with a characteristic edge length on the order of 730 nm, and thickness on the order of 125 nm. The majority of signals in the EDX spectra measured on these plates correspond to Cu and Cl, and thus the composition is indicative of copper oxychloride (Cu_2_Cl(OH)_3_), which has been identified in the sample analysis via Raman spectroscopy and XRD.

By contrast, the predominant morphology in the red sample (ChR) was smoother than that in the green sample. The strongest signals in the EDX spectra correspond to Fe and Ti, which are consistent with the chemical compositions iron and titanium oxides determined via the other techniques performed on this sample. In addition, a number of crystals of different sizes were observed, among which Al and Si predominate in some cases, and calcium in other cases.

The blue sample (ChB) shows a granular morphology with rounded holes and few differentiated crystals. The EDX spectrum of the predominant morphology shows majority of the signs of Na, Ca, Si, Al and S, consistent with the presence of lazurite in the sample. On the other hand, the crystals present a predominant signal of Ca, which indicates the calcite phase CaCO_3_, the only crystalline phase identified via XRD. 

Finally, the predominant morphology in the white sample (ChW) was relatively smooth and had rounded holes. The composition obtained via EDX comprises Ti, Si and Al as the majority elements. Similar to those for the previous samples, the most intense signal of this EDX spectrum indicates the presence of Ca, which is associated with the calcite phase.

Samples ChR and ChW show Al and Si signals in agreement with the kaolinite signals identified by FTIR.

## 5. Discussion

The mural painting is characterised from the analysis of the available samples, located at the base of an arch inside the chapel and composed of several overlaid layers of different pigments. The overlapping of the layers may have also been performed on the pointed arch that frames access to the chapel, on which the small green points represented by the ChG sample were observed. This may explain the presence of different compounds, such as clinoatacamite, atacamite, azurite and malachite, in this sample. However, clinoatacamite and atacamite may have also been produced from azurite and malachite decomposition, which is a common phenomenon [51]. Malachite commonly occurs as a mineral alongside azurite. The presence of these pigments indicates that the mural was created probably before the end of the 18th century, because since then, these pigments have disappeared, to be replaced by more stable pigments and those with synthetic components [52]. Furthermore, a study of the north façade of the Cathedral of Santo Domingo found that another green sample was made from malachite partially altered in chlorinated copper carbonates, with small inclusions of ochre earth rich in clays and iron oxides. The green pigment grains were immersed in a gypsum matrix [53].

The red sample (ChR) is representative of iron compounds widely used as pigments of different shades ranging from yellow to red. Through mineralogical characterisation techniques, an iron oxide in the form of hematite (Fe_2_O_3_) was identified with a small proportion of TiO_2_ (anatase), a white pigment that may have been added to lighten the colour or may have originated from a previous painting. Specifically, this hematite is of ferruginous clay, a product of the weathering of limestone rock. This type of ferruginous soil is very common in the colonial city of Santo Domingo and is generally located at a shallow depth between two layers of limestone or on top of limestone. In archaeological studies carried out in the Cathedral of Santo Domingo at a depth of 2 m, reddish soil was found [54]. According to chemical analyses carried out in two bricks samples using X-ray fluorescence, the presence of silicon oxide (SiO_2_), aluminium oxide (Al_2_O_3_), ferric oxide (Fe_2_O_3_), calcium oxide (CaO), magnesium oxide (MgO), sodium oxide (Na_2_O), titanium oxide (TiO_2_) and other minerals was determined. In the case of ferric oxide (Fe_2_O_3_), the two clay samples showed 10.32% and 10.60% [55].

On the other hand, titanium oxide provides colour stability, radiation reflection and durability, and at the same time protects the painted surface. Additionally, while iron oxide has been used since ancient times as a pigment, anatase is a relatively modern pigment, so that it may be inferred that this layer of paint corresponds to a recent repair action.

The compositional characterisation confirms the presence of Fe, Ti and O in the sample, with significant amounts of Al, Si or Ca, suggesting a mixture of iron and titanium oxides with other compounds. Anatase is not used as a mineral, but rutile is, so the presence of kaolinite in the rutile sample is justified [56]. This observation confirms the use of a natural pigment. 

The mixture may have included a binder that provided cohesiveness and fixed the pigment, which would indicate the use of tempera. In the tempera technique, the pigments are applied onto a dry lime plaster and are fixed using an organic binder. Superposition of the binder and pigments applied onto the plaster allows the paint to penetrate slightly into the plaster. 

Almost all technical characteristics of a painting, including texture, drying speed, adherence, optical qualities, resistance and opacity or transparency of the pigments, depend on the binder. Among the different types of binders used until the end of the 18th century, tempera was of animal or vegetable origin, such as glue or gum, blood serum, animal fat, casein and vegetable resin. Sand, clay and other materials were also used as additives [57]. In Santo Domingo, the use of blood serum or cactus like *Consolea moniliformis* commonly called *Alpargata*, has been identified. Cactus has been used for pigments since the 16th century, specially the Cochineal (*Grana cochinilla*). Other studies have determined that the nopal (*Opuntia ficus-indica*) mucilage was used as well in construction in the colonial time [58]. In this study, none of the samples revealed the presence of any type of those binders, but the presence of calcite and gypsum. Taking into account this result, it may be inferred that the mural painting analysed is a fresco.

The reflectance in the UV–VIS range of the blue sample (ChB) suggests that the pigment used was ultramarine blue, which originates from the mineral lapis lazuli [33]. The generic composition of this pigment is ((Na,Ca)_8_(AlSiO_4_)_6_(SO_4_,S,Cl)_2_), with fixed proportions of aluminium, silicon and oxygen in an aluminosilicate framework and variable proportions of the other elements. A compositional analysis of the predominant granular morphology in the sample revealed that majority of its elements are Na, Al, Si, S, Ca and O. This composition is consistent with that of lazurite, Na_3_Ca(Si_3_Al_3_)O_12_S, which was identified via Raman spectroscopy and is the main component of lapis lazuli. In addition, both XRD and SEM–EDX detected the presence of calcite, which is a typical minority component of natural lapis lazuli. 

Lazurite is a stable pigment that can be used in aqueous paint, in egg tempera or vegetable gums and in oily media. Due to its stability against alkalis, it is also compatible with lime. It works best with tempering binders. The presence of lazurite in the chapel, which is provided by the ultramarine-blue natural pigment (lapis lazuli), indicates that the mural painting in the Our Lady of Mercy chapel was created between the 17th and 18th centuries. Although this pigment was recently reported to have been in use since antiquity [43], its use in European painting decayed since the middle of the 18th century. Lazurite is a very stable pigment that provides permanence of colour. Because of its properties and the price of lapis lazuli powder, it was referred to as ‘blue gold’.

Finally, the white sample (ChW) has a reflectance spectrum characteristic of titanium-oxide-based pigments, which must also be mixed with other compounds, because the predominantly smooth morphology exhibits significant contributions from Al, Si and Ca, in addition to Ti and O. The mineralogical composition of this sample, determined via XRD and Raman spectroscopy, confirmed the presence of titanium oxide (TiO_2_) in the rutile phase as the fundamental component of the white pigment. Since it also has kaolinite, it may indicate that it comes from the rutile mineral.

The compositional analysis of all samples revealed in some areas a clear predominance of Ca and O atoms, whereas through Raman spectroscopy, the presence of calcite was identified in the backs of the samples, confirming that successive layers of paint were applied over a lime base.

## 6. Conclusions

Through optical, chemical, mineralogical and microstructural analyses, a mural painting in a 17th-century chapel of the first cathedral of the Americas, located in the colonial city of Santo Domingo, was characterised. The presence of four pigments, i.e., green, red, blue and white, was determined based on colours visually observed in different areas of the mural painting. Samples of the four pigments were analysed using reflectance spectra in the UV–VIS–NIR range, both in situ and in the laboratory. 

From the spectra, the colorimetric properties of the samples were quantified in terms of coordinates in the CIELAB space (a*, b* and L*). Optical characterisation in situ provided coherent information on the variety of colours and shades identified. The average colour coordinates obtained in the laboratory constitute a quantitative characterisation of the predominant colours in the samples of mural paintings located in the Our Lady of Mercy Chapel of the first cathedral of the Americas.

The chromatophore of the green pigment was probably malachite, which is prone to reactions with chlorides in the environment that may have led to the presence of copper oxychloride in the painting. The blue pigment is lazurite, which is the main component responsible for its ultramarine-blue pigment. The ochre pigment originated from the earth, whereas the red pigment is a calcined ochre. In addition, the first layer observed in the samples was determined to have been applied directly onto the stone support, and above that layer, several layers were overlapped. This layering indicates that the chapel had been repainted at different times with different pigments throughout its history. 

In summary, the characterisation of a mural painting in situ via non-destructive techniques, complemented with analyses in the laboratory, allows clear and precise information to be obtained about the compositions of its samples. According to its characterisation, the mural painting examined in this study was created before the end of the 18th century, because many of its materials were no longer used after the industrialisation of painting.

## Figures and Tables

**Figure 1 materials-14-06866-f001:**
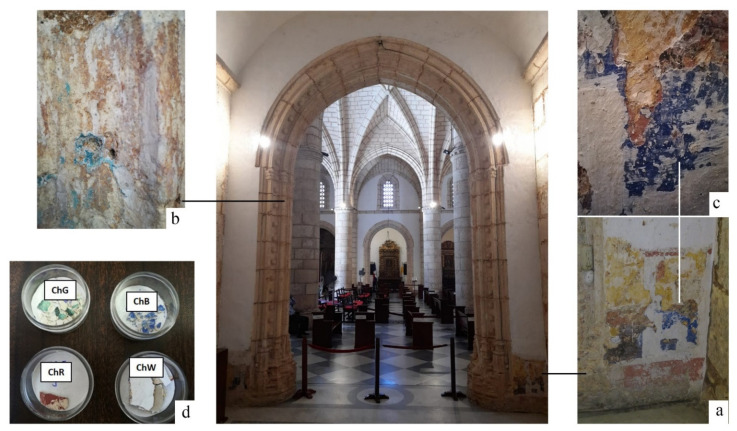
Remains of mural painting of Our Lady of Mercy Chapel; (**a**–**c**) zones from which the samples were taken; (**d**) analysed samples.

**Figure 2 materials-14-06866-f002:**
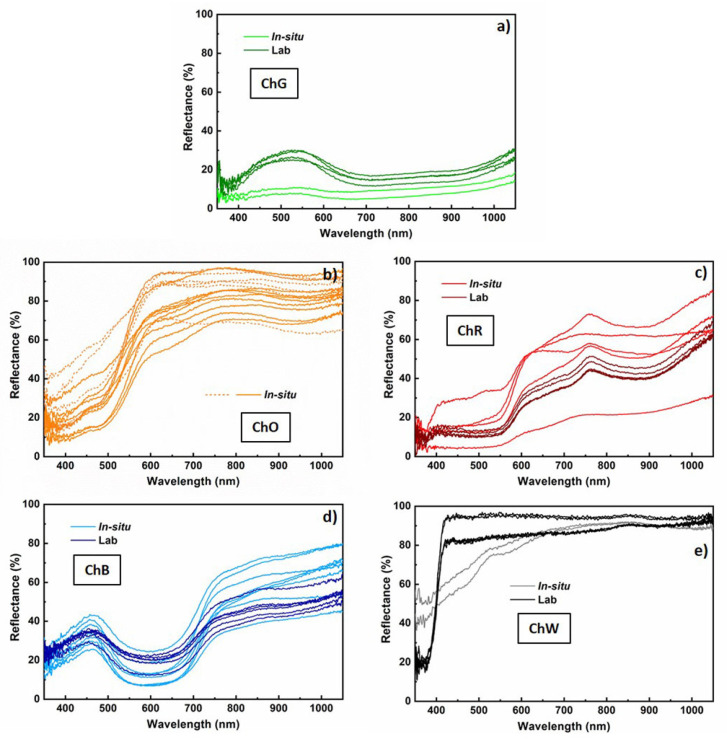
Reflectance spectra obtained from mural painting both in situ and in the laboratory. (**a**) green (ChG); (**b**) ochre (ChO); (**c**) red (ChR); (**d**) blue (ChB) and (**e**) white (ChW).

**Figure 3 materials-14-06866-f003:**
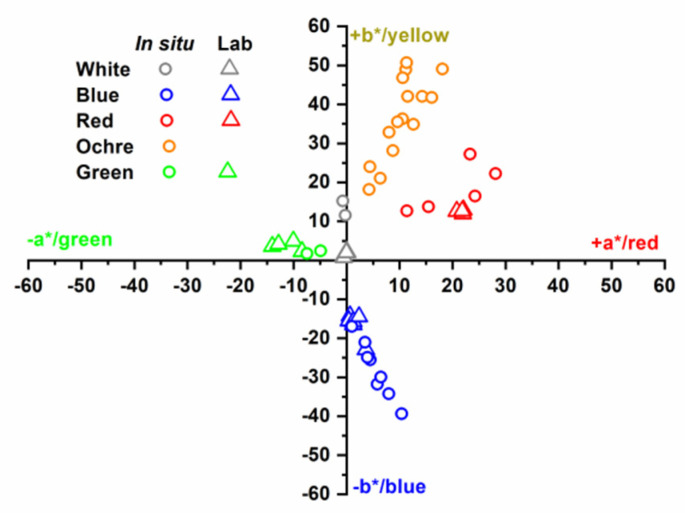
Colour coordinates of mural painting.

**Figure 4 materials-14-06866-f004:**
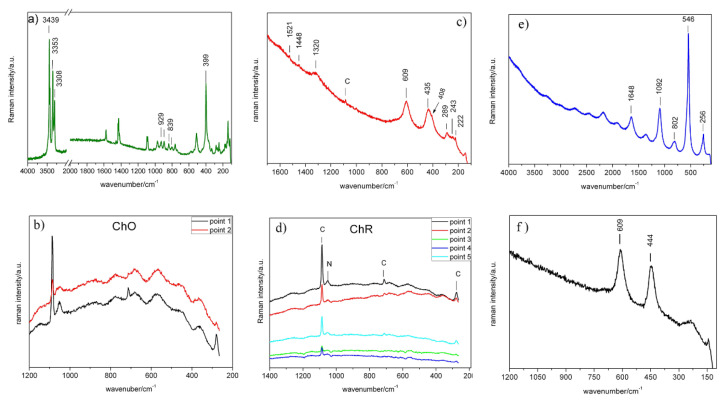
Raman spectra of the samples. Green sample (ChG) in laboratory (**a**) Ochre sample (ChO) in situ (**b**); red sample (ChR) in laboratory (**c**) and in situ (**d**); blue sample (ChB) in laboratory (**e**) and white sample (ChW) in laboratory (**f**).

**Figure 5 materials-14-06866-f005:**
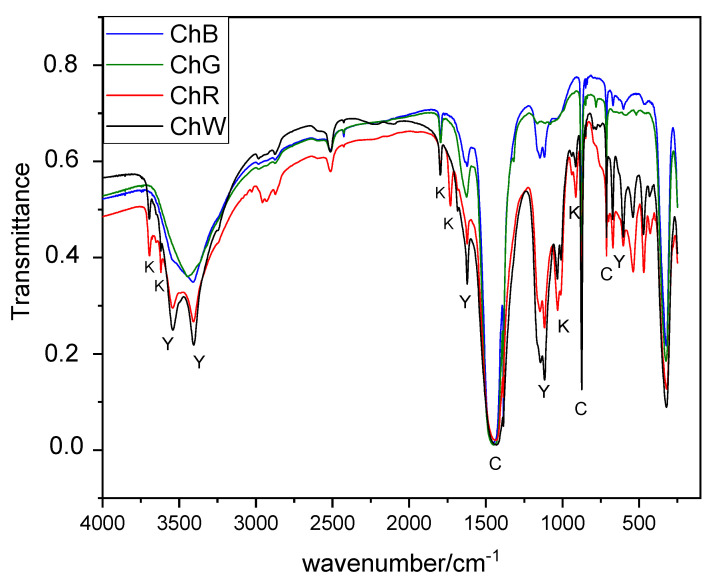
FTIR results of samples: C = calcite (CaCO_3_); Y = gypsum (CaSO_4_·2H_2_O); K = kaolinite (Al₂Si₂O₅(OH)₄).

**Figure 6 materials-14-06866-f006:**
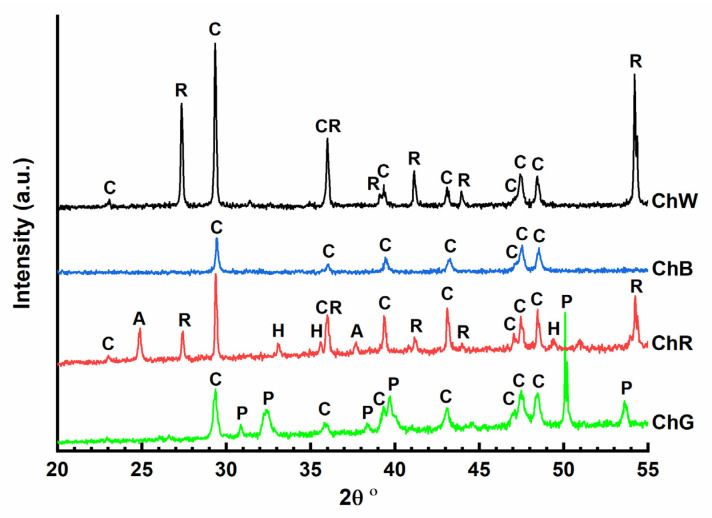
X-ray diffraction results of samples: C-calcite (CaCO_3_); R-rutile (TiO_2_); A-anatase (TiO_2_); H-hematite (α-Fe_2_O_3_); P-paratacamite (Cu_2_Cl(OH)_3_).

**Figure 7 materials-14-06866-f007:**
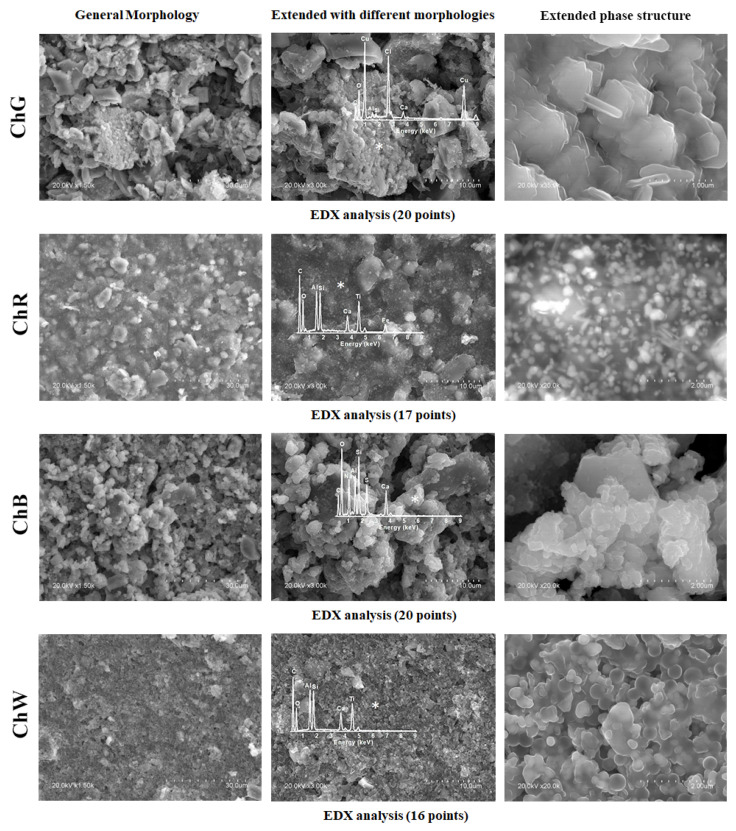
Microstructural analysis of samples via SEM/EDX.

**Table 1 materials-14-06866-t001:** Chemical and mineralogical characterisation of the samples.

Sample		UV-vis-NIR	Raman	FTIR	XRD	SEM/EDX
ChG	In situ		Calcite	Calcite, gypsum, anhydrite	Calcite, paratacamite	Cl, Cu, Ca
	Lab	Atacamite related compounds (Cu_2_Cl(OH)_3_	Azurite, Cu_2_Cl(OH)_3_			
ChR	In situ	α-Fe_2_O_3_	Calcite, Nitrates	Calcite, gypsum; kaolinite; αFe_2_O_3_	Calcite; αFe_2_O_3_; anatase; rutile	Fe, Ti, Si, Al, Ca
	Lab	α-Fe_2_O_3_	α-Fe_2_O_3_; TiO_2_ (rutile and anatase); CacO_3_.			
ChB	In situ	Ultramarine blue(Na,Ca)_8_(AlSiO_4_)_6_(SO_4_,S,Cl)_2_		Calcite, gypsum; anhydrite	Calcite	Na, Ca, Si, Al
	Lab	Ultramarine blue (Na,Ca)_8_(AlSiO_4_)_6_(SO_4_,S,Cl)_2_	Lazurite (Na_3_Ca(Si_3_Al_3_)O_12_S			
ChW	In situ			Calcite, gypsum; kaolinite, rutile	Calcite, rutile	Ti, Si, Al, Ca
	Lab	TiO_2_	TiO_2_ (rutile)			
ChO	In situ	FeO(OH)·nH_2_O; α-FeO(OH)		n.d	n.d	n.d
	Lab	n.d	n.d			

n.d. not determined.

**Table 2 materials-14-06866-t002:** Colour coordinates of the mural pigments.

Colour		L*	a*	b*
**Green**	In-situ	34.8 ± 4.4	−6.2 ± 1.8	2.1 ± 0.5
	Lab-ChG	57.2 ± 2.2	−11.3 ± 2.6	3.8 ± 1.1
**Ochre**	In-situ	80.2 ± 6.9	10.5 ± 3.9	36.8 ± 10.4
**Red**	In-situ	54.9 ± 13.7	20.5 ± 6.9	18.5 ± 6.1
	Lab-ChR	48.1 ± 2.2	21.7 ± 0.5	12.6 ± 0.4
**Blue**	In-situ	46.3 ± 8.0	5.5 ± 2.9	−28.0 ± 7.3
	Lab-ChB	53. ± 4.0	1.6 ± 1.2	−16.6 ± 3.2
**White**	In-situ	90.5 ± 1.3	−0.5 ± 0.4	13.4 ± 2.,6
	Lab-ChW	95.1 ± 2.,3	−0.2 ± 0.3	1.7 ± 0.7

## Data Availability

Not applicable.

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
