# Peer review of "Physical and Chemical Characterisation of the Pigments of a 17th-Century Mural Painting in the Spanish Caribbean"

_materials, 2021, doi:10.3390/ma14226866_

Round 1
Reviewer 1 Report
The article “Physical and chemical characterization of the pigments of a 17th century wall painting in the Spanish Caribbean” concerns a physical and chemical characterization of the pigments used on a mural painting in the Spanish Caribbean. In particular, the mural painting is located in a 17th century-chapel of the Cathedral of Santo Domingo. The authors use a multi analytical approach to characterize the pigments, in order to obtain information on the history of the building. The authors establish that the mural painting was realize before the end of the 18th century on the basis of the obtained results. Specifically, they considered that many of the characterized pigments are no longer used after industrialization, except for the presence of TiO2, that they attribute to a subsequent restoration. The object of the analyses could be potentially interesting from a historical and artistic point of view but there are several adjustments and details that the authors should be added before to take in consideration this work for the publication. Firstly, a valid dissertation must be added in order to justify how the authors are able to date that the painting was realized before the end of the 18th century. As a matter of fact, the assumption that some of the characterized pigments were no longer used after industrialization is not sufficient. Many pigments, such as malachite and lapis lazuli, continued to be used by the artists even in the 19th century and the beginning of the 20th century. How can authors date a painting only on the basis of the pigments found? The related motivation must be strengthened, otherwise it would be considered as not valid. Then, it is not clear what the authors want to characterize: are they referring to the pigments, the technique or the historical period? They focus the whole work on the characterization of pigments but they did not considered the binder. The characterization of the binder is mandatory in order to obtain information about the technique used by the artist, in my opinion. This aspect should be discussed further and supported by additional measurements, if possible. In addition, it is not clear the number of samples analyzed and in which form they were analyzed. Finally, the use of the word "color" in the text is incorrect and , should be replaced with "pigments or dyes" or “hue”.
My opinion is that the work could be reconsider after major revision.
Following Comments and Suggestions for Authors.
English must be revised.
Abstract:
In my opinion it is really difficult, or even impossible, to establish that a mural painting was completed before the end of the 18th century only because some pigments were no longer in use after industrialization. The argument should be developed further. Many pigments, such as malachite, lapis lazuli and ocher, continued to be used by the artists even in the 19th century and at the beginning of the 20th. How can authors date a painting with certainty only on the basis of the pigments found? The motivation must be strengthened, otherwise it would be considered as not valid. If the authors are sure that TiO2 is due to a restoration they will be required to prove it, or consider that it may have been used directly by the artist or present as an additive in modern protective products.
Introduction:
- Rows 32-33: “However, lack of maintenance, abandonment, and oblivion have caused much of them to disappear, leaving only traces”. The factors mentioned by the authors could certainly favor the degradation of the painted surfaces but it is not clear what kind of artwork they refer to. Are they referring to paintings? Or Frescoes? Or Mural Painting? The degradation depending on the type of materials. These aspects should be clarified in the text. Replace "lack of maintenance" with "degradation process".
- Rows 34-35: “When pigments with different colours”. In my opinion, the problem is not related to the use of different pigments but rather to the fact that they have a different chemical composition. In addition, the following aspects have to be considered: their interaction on the paint layer, the interaction between pigments and binders, the substrate and the conservation The use of the word "color" in the text is incorrect and should be replaced with "pigments or dyes" or “hue”.
- Rows 35-36: It is not clear how the pigments / dyes used can give information on the historical period. This aspect should be clarified.
- Rows: 36- 37 : “With the recovery of colours, we can date buildings and spaces, but before that can be accomplished, it is necessary to know the properties of these colours.” The authors say that by characterizing the “color” (the word "color" should be replaced with pigments or dyes, in my opinion) it is possible to date the buildings. However it is not correct since an exact dating of the pigments and dyes used is almost impossible with the proposed techniques. Surely, it is possible to obtain information about the period in which some pigments were used. But nothing more than that. The sentence should be rephrased accordingly.
- Rows 38-39: “The study of pigments provides complementary information regarding architecture, culture, and society. In the field of historical heritage, the characterisation of colour allows the study of the historical evolution of a building and relates it to technical, constructive, aesthetic, and social aspects”. The authors should explain how the use of certain pigments helps to provide complementary information on the history of the building.
- Rows 41-44: “In addition, the study of colour provides information on the degradation processes and maintenance and repair actions that the materials have undergone over time. For these reasons, colour characterisation has become increasingly important for the restoration and maintenance of historical buildings.” The authors say that the color characterization produces information on the conservation state of the analyzed building but it is not sufficiently clear. The relationship between pigment degradation and buildings has to be explained. Are they frescoes? Or Mural paintings? In which period were they realized? These aspects are not clear and there is no inherent bibliography. Furthermore, the pigment-binder mixture, i.e. the paint layer, has to be characterized, rather than the "color".
- Rows: 57-64: “colour acquisition may be influenced by the effects of environment on outdoor measurements, by the morphology of a studied object with concavities and convexities, which complicate the analysis, or by material changes”. What do the authors mean? Why should an in situ measurement be disturbed compared to one made in the laboratory? This comparison is not clear and should be explained. The authors discuss about color studies but they do not specify in what terms. What does color analysis mean? Does it refer to the colorimetry analysis? FORS analysis? Or is it simply a stylistic analysis? In addition, the authors have to clarify what they exactly mean by the word "color". Do they refer to pigments? Or to the pictorial layer? It is not clear.
- Rows 66- 75: The authors refer about analyzes conducted on pigments. However it is not clear which is the subject of the analyzes. Please clarify.
- Rows 80: Please explain how is it possible “to determine their colour…” by using SEM-EDX analyses.
- Rows 133- 136: The entire methodological approach should be better discussed.
I believe that the introduction should be modify in order to clarify the purpose of the work and the results obtained.
- Our Lady of Merci Chapel in the Chatedral of Santo Domingo:
- Row 157-160: Please indicate the points that have been analyzed. Has any sampling been made? How were the pigments analyzed? Were they embedded in thin sections? The methodologies description should be improved.
- Materials and Methods:
- Rows 163-170: The authors should explain better the number of samples analyzed, how they were sampled, and if stratigraphic sections were made. If the authors have photos taken with an optical microscope, it would be useful to include them in order to further discuss the morphology of the pigments.
- Rows 200-201: The authors say that ocher cannot be transported in laboratory because it is applied directly to the stone. What does this mean, exactly? Is it a fresco? How were the other pigments applied? It should be specified better.
- Row 203-209: The authors repeated the reflectance measurements in the laboratory. Could you please clarify the reason? It should be specified.
- Two different Raman instruments were used by the authors. Why? Does the choice lie in the excitation wavelength?
- Results:
4.1 In situ characterization
- Row 222: Replace “remains” with “sample”.
- Row 300-301: Please give a plausible reason why Raman does not give a signal. Is there fluorescence? Considering that both lapis and hematite have a Raman signal, could this issue be due to the binder? Why the authors preferred to carry out destructive analyzes in the laboratory (SEM-EDX) instead of using a portable XRF? For the identification of the binder, infrared spectroscopy (FT-IR) could be more useful instead of Raman.
4.2. Characterization of samples in the laboratory:
4.2.1 Optical Characterization
- In my opinion, the authors should specify what the laboratory results add with respect to the in situ ones. It is not clear.
4.2.2 Raman spectroscopy
- It is not clear how the samples have been analyzed. Have stratigraphic sections been made? Were they embedded in resin?
- Rows 364-366: "Breakdown" should be replaced with "degradation". The authors argue that the degrade is due to chlorides which are present in the surrounding environment. How can this statement be demonstrable?
- Rows 367-368: Do the authors claim that azurite is due to a process of malachite degradation? They should motivate it. Why they assume that it is a degradation and not a pigment directly used by the artist?
4.2.4 X-ray diffraction (XRD)
- The name of the paragraph is wrong, should be SEM-EDX.
- How do the authors explain the presence of Al and Si in the white titanium pigment?
- Discussion:
- Rows 458-460: “Malachite commonly occurs as a mineral alongside azurite and is used as a hardening binder. The presence of these pigments indicates that the mural was created before the end of the 18th century, because since then, these pigments have disappeared, to be replaced by more stable pigments and those with synthetic components”. What does it mean that malachite is used as a hardening binder? It is a pigment. The authors must explain this phrase or add bibliography. Moreover, the characterization of malachite as a prove to date a mural painting is very week as a thesis. It should be better supported. Malachite can also be found on 19th an 20th century paintings.
- Rows 463-466: “The red sample (ChR) is representative of iron compounds widely used as pigments of different shades ranging from yellow to red. Through mineralogical characterisation techniques, an iron oxide in the form of hematite (Fe2O3) was identified with a small proportion of TiO2, a white pigment that may have been added to lighten the colour or may have originated from a previous painting.” The authors affirm that TiO2 may be due to an earlier painting and they date the painting to the end of the 18th century. However, if TiO2 was discovered in 19th century how it would be possible that is due to a previous painting? In my opinion they should better support this result.
- Rows 464-471: “have been added to lighten the colour or may have originated from a previous painting. Specifically, this hematite is of ferruginous clay, a product of the weathering of limestone rock. This type of ferruginous soil is very common in the colonial city of Santo Domingo and is generally located at a shallow depth between two layers of limestone or on top of limestone. On the other hand, titanium oxide provides colour stability, radiation reflection, and durability, and at the same time protects the painted surface.” It is not clear what the authors mean since there is no supporting bibliography.
- Rows 475-479: “The mixture may have included a binder that provided cohesiveness and fixed the pigment, which would indicate the use of tempera. In the tempera technique, the colours are applied onto a dry lime plaster and are fixed using an organic binder. Superposition of the binder and pigments applied onto the plaster allows the paint to penetrate slightly into the plaster. In quality work, the plaster is fine. On the other hand, if no binder is used, the technique is referred to as fresco.” Further analyzes should be made to identify the binder and making a stratigraphic section could help to understand if it is a fresco or a mural painting. The different conservation of the two techniques can also provide information. The presence of calcite could indicate the use of fresco.
- Rows 490-491: “In this study, none of the samples revealed the presence of any type of binder, perhaps because of low concentrations or the passage of time.” This sentence is not clear. How do the authors establish that the binders are not present? If the pigment is still cohesive onto the stone, it implies that the binder is still there. It is the binder that allows the pigments to be cohesive to the surface. It must be specified. The authors found calcite. Why don't they hypothesize the use of fresco?
The discussions should be improved in order to explain how the results obtained make it possible to date the mural painting before the end of 18th century.

Author Response
Thank you for your time and valuable and helpful comments. We have analyzed and revised our manuscript according to your remarks. All changes are indicated in the text. In addition, we send the unified version and version with tracked changes, as the editor requires and our responses below.
The article “Physical and chemical characterization of the pigments of a 17th century wall painting in the Spanish Caribbean” concerns a physical and chemical characterization of the pigments used on a mural painting in the Spanish Caribbean. In particular, the mural painting is located in a 17thcentury-chapel of the Cathedral of Santo Domingo. The authors use a multi analytical approach to characterize the pigments, in order to obtain information on the history of the building. The authors establish that the mural painting was realize before the end of the 18th century on the basis of the obtained results. Specifically, they considered that many of the characterized pigments are no longer used after industrialization, except for the presence of TiO2, that they attribute to a subsequent restoration. The object of the analyses could be potentially interesting from a historical and artistic point of view but there are several adjustments and details that the authors should be added before to take in consideration this work for the publication. Firstly, a valid dissertation must be added in order to justify how the authors are able to date that the painting was realized before the end of the 18th century. As a matter of fact, the assumption that some of the characterized pigments were no longer used after industrialization is not sufficient. Many pigments, such as malachite and lapis lazuli, continued to be used by the artists even in the 19th century and the beginning of the 20th century. How can authors date a painting only on the basis of the pigments found? The related motivation must be strengthened, otherwise it would be considered as not valid. Then, it is not clear what the authors want to characterize: are they referring to the pigments, the technique or the historical period? They focus the whole work on the characterization of pigments but they did not considered the binder. The characterization of the binder is mandatory in order to obtain information about the technique used by the artist, in my opinion. This aspect should be discussed further and supported by additional measurements, if possible. In addition, it is not clear the number of samples analyzed and in which form they were analyzed. Finally, the use of the word "color" in the text is incorrect and , should be replaced with "pigments or dyes" or “hue”.
Thank you very much for your valuable comments, which will be answered below and modified in the text.
My opinion is that the work could be reconsider after major revision.
Following Comments and Suggestions for Authors.
English must be revised.
Abstract:
In my opinion it is really difficult, or even impossible, to establish that a mural painting was completed before the end of the 18th century only because some pigments were no longer in use after industrialization. The argument should be developed further. Many pigments, such as malachite, lapis lazuli and ocher, continued to be used by the artists even in the 19th century and at the beginning of the 20th. How can authors date a painting with certainty only on the basis of the pigments found? The motivation must be strengthened, otherwise it would be considered as not valid. If the authors are sure that TiO2 is due to a restoration they will be required to prove it, or consider that it may have been used directly by the artist or present as an additive in modern protective products.
Introduction:
- Rows 32-33: “However, lack of maintenance, abandonment, and oblivion have caused much of them to disappear, leaving only traces”. The factors mentioned by the authors could certainly favor the degradation of the painted surfaces but it is not clear what kind of artwork they refer to. Are they referring to paintings? Or Frescoes? Or Mural Painting? The degradation depending on the type of materials. These aspects should be clarified in the text. The degradation depending on the type of materials. These aspects should be clarified in the text. We are considering a plaster an it is included in the text. Replace "lack of maintenance" with "degradation process". The sentence has been rewritten, since the degradation process can be produce by lack of maintenance.
- Rows 34-35: “When pigments with different colours”. In my opinion, the problem is not related to the use of different pigments but rather to the fact that they have a different chemical composition. In addition, the following aspects have to be considered: their interaction on the paint layer, the interaction between pigments and binders, the substrate and the conservation environment.
We agree that it is not the pigment but its chemical composition and the sentence has been rewritten
The use of the word "color" in the text is incorrect and should be replaced with "pigments or dyes" or “hue”.
This have been changed. However, one of the techniques used is a colorimeter and therefore the colour of the samples is measured. In these cases the word “color” has been kept.
- Rows 35-36: It is not clear how the pigments / dyes used can give information on the historical period. This aspect should be clarified.
This paragraph refers for example to the fact that pigments have different prices (cheap and expensive). Some come from the Spanish peninsula and others from the New World. This helps to identify if the pigment was local or imported from Europe. Fashion, usage, and custom help determine an era and social status. This helps to determine the social aspect and to know if that fashion came to the island or not.
- Row: 36- 37 : “With the recovery of colours, we can date buildings and spaces, but before that can be accomplished, it is necessary to know the properties of these colours.” The authors say that by characterizing the “color” (the word "color" should be replaced with pigments or dyes, in my opinion) it is possible to date the buildings. However it is not correct since an exact dating of the pigments and dyes used is almost impossible with the proposed techniques. Surely, it is possible to obtain information about the period in which some pigments were used. But nothing more than that. The sentence should be rephrased accordingly.
Thank you for the observation. We agree with the reviewer, with these techniques we cannot date the pigments. But we can get information about the period and techniques in which pigments were used as you said. The sentence has been rewritten.
- Rows 38-39: “The study of pigments provides complementary information regarding architecture, culture, and society. In the field of historical heritage, the characterisation of colour allows the study of the historical evolution of a building and relates it to technical, constructive, aesthetic, and social aspects”. The authors should explain how the use of certain pigments helps to provide complementary information on the history of the building.
The sentence has been rewritten to clarify.
As mentioned in a previous point: This paragraph refers for example to the fact that pigments have different prices (cheap and expensive). Some come from the Spanish peninsula and others from the New World. This helps to identify if the pigment was local or imported from Europe. Fashion, usage, and custom help determine an era and social status. This helps to determine the social aspect and to know if that fashion came to the island or not.
- Row 41-44: “In addition, the study of colour provides information on the degradation processes and maintenance and repair actions that the materials have undergone over time. For these reasons, colour characterisation has become increasingly important for the restoration and maintenance of historical buildings.” The authors say that the color characterization produces information on the conservation state of the analyzed building but it is not sufficiently clear. The relationship between pigment degradation and buildings has to be explained. Are they frescoes? Or Mural paintings? In which period were they realized? These aspects are not clear and there is no inherent bibliography. Furthermore, the pigmentbinder mixture, i.e. the paint layer, has to be characterized, rather than the "color".
Thank you for the observation. We agree with the reviewer. It seems that the original idea was lost during translation, which is “the study of colour provides information for maintenance and preservation of the buildings”. The sentence has been rewritten
- Rows: 57-64: “colour acquisition may be influenced by the effects of environment on outdoor measurements, by the morphology of a studied object with concavities and convexities, which complicate the analysis, or by material changes”. What do the authors mean? Why should an in situ measurement be disturbed compared to one made in the laboratory? This comparison is not clear and should be explained. The authors discuss about color studies but they do not specify in what terms. What does color analysis mean? Does it refer to the colorimetry analysis? FORS analysis? Or is it simply a stylistic analysis? In addition, the authors have to clarify what they exactly mean by the word "color". Do they refer to pigments? Or to the pictorial layer? It is not clear.
This paragraph refers to the fact that time and environmental conditions can affect the pigments and change its hue. Also, depending on where the pigments are located (especially for in situ) makes the analysis more difficult. The paragraph has been modified.
- Rows 66- 75: The authors refer about analyzes conducted on pigments. However it is not clear which is the subject of the analyzes. Please clarify.
The sentence has been rewritten to clarify.
- Rows 80: Please explain how is it possible “to determine their colour…” by using SEMEDX analyses.
We agree with the reviewer, optical microscopy was used to determine the colour and SEM to determine the microstructure. Through EDX the elemental composition of the pigments was determined. This was explained in the document.
- Rows 133- 136: The entire methodological approach should be better discussed. I believe that the introduction should be modify in order to clarify the purpose of the work and the results obtained.
Thank you for the suggestion. New explanation about the methodology has been added.
- Our Lady of Merci Chapel in the Chatedral of Santo Domingo:
- Row 157-160: Please indicate the points that have been analyzed. Has any sampling been made? How were the pigments analyzed? Were they embedded in thin sections? The methodologies description should be improved.
We have modified Figure 1, adding a general photo of the chapel to indicate the location of the samples. An explanation has been added.
- Materials and Methods:
- Rows 163-170: The authors should explain better the number of samples analyzed, how they were sampled, and if stratigraphic sections were made. If the authors have photos taken with an optical microscope, it would be useful to include them in order to further discuss the morphology of the pigments.
The paragraph has been rewritten explaining better the samples analyzed.
Unfortunately we do not have photos from an optical microscope.
- Rows 200-201: The authors say that ocher cannot be transported in laboratory because it is applied directly to the stone. What does this mean, exactly? Is it a fresco? How were the other pigments applied? It should be specified better.
The sentence has been rewritten to clarify.
- Row 203-209: The authors repeated the reflectance measurements in the laboratory. Could you please clarify the reason? It should be specified.
- The importance of the laboratory reflectance measurements was highlighted in the results section for each sample. In the revised version of the text, the details for all the samples have been collected together in the Materials and methods section, in order to better clarify this importance: “This analysis is expected to give rise to more accurate reflectance spectra with respect to the on-site one, as flat and clean areas of the samples were selected to ensure optimal measurement conditions. More specifically, four spectra were measured at different points of the largest green fragment (ChG) which had a non-homogeneous green colour representative of the predominant shades in the re-mains, lighter than in the area measured in situ. For the red sample (ChR), spectra were obtained at five different points within the flattest, darkest, and most uniform parts. Two different small fragments were measured for the blue sample (ChB), and three spectra were obtained at different points within each fragment. Finally, the white outer layer (ChW) was characterised more reliably in the laboratory than in situ, as the cleanest and flattest areas of the samples were measured at five different points.” Besides, the on-site characterization provides a more complete information on the variety of colours and shades that were identified in the painting, as well as the analysis of the ochre layer. Consequently, the authors believe that the two sets of measurements were necessary.
- Two different Raman instruments were used by the authors. Why? Does the choice lie in the excitation wavelength?
Two different Raman instruments were used because one is portable and the other one is in the laboratory, since the resolution of the portable one is lower that the laboratory one and no always is possible to get good spectra with the portable equipment. Additionally, the laser wavelength is different in both equipment in order to perform a better characterization of the samples. The portable instrument has a 785 nm laser that avoids fluorescence, but lower resolution than the laboratory instrument. In the laboratory instrument the laser has a frequency of 532 nm.
- Results:
4.1 In situ characterization
- Row 222: Replace “remains” with “sample”.
It has been replaced here and in other parts of the manuscript.
Row 300-301: Please give a plausible reason why Raman does not give a signal. Is there fluorescence? Considering that both lapis and hematite have a Raman signal, could this issue be due to the binder?
There were no Raman signals due to the low resolution of the portable equipment. In addition, if the absence of Raman signals were due to the binder, there would also be no signal with the laboratory equipment.
Why the authors preferred to carry out destructive analyzes in the laboratory (SEM-EDX) instead of using a portable XRF?
We agree with the referee that XRF would be sufficient to obtain information on the elemental chemical composition of the sample, however we preferred to do also SEM/EDX, since we can obtain information on its morphology. On the other hand, the sample had already been detached from the support and had been used for other analyses. The amount of sample used in the SEM/EDX analysis was about 2-5 mm.
For the identification of the binder, infrared spectroscopy (FT-IR) could be more useful instead of Raman.
FTIR of the samples in KBr pellets have been made in all the samples and the information is included in the manuscript.
4.2. Characterization of samples in the laboratory:
4.2.1 Optical Characterization
In my opinion, the authors should specify what the laboratory results add with respect to the in situ ones. It is not clear.
The answer to this comment has been detailed before (comment for rows 203-209). As explained then, the authors consider that the two sets of results are complementary and necessary. The on-site results provide the characterization of the ochre layer and of the various colours observed in the remains, while the laboratory ones provide a more accurate characterization of the predominant colour of each layer. The latter is especially evident in the green and white remains in which the in-situ analysis was affected by their location and pollution, respectively.
4.2.2 Raman spectroscopy
It is not clear how the samples have been analyzed. Have stratigraphic sections been made? Were they embedded in resin?
The Raman spectra of the samples were performed directly on the samples, without any type of treatment, they were not embedded in any resin and no stratigraphy was performed. The samples of the different pigments were selected and placed directly on the sample holder of the Raman spectrophotometer.
A paragraph explaining this has been added.
- Rows 364-366: "Breakdown" should be replaced with "degradation". It has been replaced.
The authors argue that the degrade is due to chlorides which are present in the surrounding environment. How can this statement be demonstrable?
It should be noted that until 2008 the Cathedral was open without air conditioning. This condition allowed pollution from vehicles and industries located nearby to enter the interior of the cathedral. In 2008, air conditioning was installed, allowing better control of the interior air. Some information is included in the paragraph to clarify.
- Rows 367-368: Do the authors claim that azurite is due to a process of malachite degradation? They should motivate it. Why they assume that it is a degradation and not a pigment directly used by the artist?
Thank you for the comment. Indeed, you are right, the malachite could have been added or obtained as a degradation process of the azurite.
4.2.4 X-ray diffraction (XRD)
The name of the paragraph is wrong, should be SEM-EDX. How do the authors explain the presence of Al and Si in the white titanium pigment?
The name of the paragraph has been change.
It has been explained by FTIR, where kaolinite has been identified.
- Discussion:
- Rows 458-460: “Malachite commonly occurs as a mineral alongside azurite and is used as a hardening binder. The presence of these pigments indicates that the mural was created before the end of the 18th century, because since then, these pigments have disappeared, to be replaced by more stable pigments and those with synthetic components”. What does it mean that malachite is used as a hardening binder? It is a pigment. The authors must explain this phrase or add bibliography. Moreover, the characterization of malachite as a prove to date a mural painting is very week as a thesis. It should be better supported. Malachite can also be found on 19th an 20th century paintings.
The sentence has been changed, we agree that malaquite is not a binder, it is a pigment.
We have added the following information: ”A study of the north façade of the Cathedral of Santo Domingo found that sample CSD-4 (Green) was made from malachite partially altered in chlorinated copper carbonates, with small inclusions of ochre earth rich in clays and iron oxides. The green pigment grains are immersed in a gypsum matrix. The binder is fatty (possibly from a protein such as casein or albumin). Traces of palmitic and stearic fatty acids and a trace of azelaic acid are also present.”
- Rows 463-466: “The red sample (ChR) is representative of iron compounds widely used as pigments of different shades ranging from yellow to red. Through mineralogical characterisation techniques, an iron oxide in the form of hematite (Fe2O3) was identified with a small proportion of TiO2, a white pigment that may have been added to lighten the colour or may have originated from a previous painting.” The authors affirm that TiO2 may be due to an earlier painting and they date the painting to the end of the 18th century. However, if TiO2 was discovered in 19th century how it would be possible that is due to a previous painting? In my opinion they should better support this result.
We agree with the reviewer and probably this layer of pigment was applied recently. It has been added some comment to explain it. “Although hematite is known to have been used since ancient times as a pigment, the use of TiO2 is relatively recent, so this layer of paint corresponds to a recent repair of the chapel.”
- Rows 464-471: “have been added to lighten the colour or may have originated from a previous painting. Specifically, this hematite is of ferruginous clay, a product of the weathering of limestone rock. This type of ferruginous soil is very common in the colonial city of Santo Domingo and is generally located at a shallow depth between two layers of limestone or on top of limestone. On the other hand, titanium oxide provides color stability, radiation reflection, and durability, and at the same time protects the painted surface.” It is not clear what the authors mean since there is no supporting bibliography.
We have added the following information: “In archaeological studies carried out in the Cathedral at a depth of 2 meters, reddish soil was found. According to chemical analyses carried out in two bricks samples using X-ray fluorescence, the presence of silicon oxide (SiO2), aluminum oxide (Al2O3), ferric oxide (Fe2O3), calcium oxide (CaO), magnesium oxide (MgO), sodium oxide (Na2O), titanium oxide (TiO2) and other minerals was determined. In the case of ferric oxide (Fe2O3), the two clay samples showed 10.32% and 10.60%”
- Rows 475-479: “The mixture may have included a binder that provided cohesiveness and fixed the pigment, which would indicate the use of tempera. In the tempera technique, the colours are applied onto a dry lime plaster and are fixed using an organic binder. Superposition of the binder and pigments applied onto the plaster allows the paint to penetrate slightly into the plaster. In quality work, the plaster is fine. On the other hand, if no binder is used, the technique is referred to as fresco.” Further analyzes should be made to identify the binder and making a stratigraphic section could help to understand if it is a fresco or a mural painting. The different conservation of the two techniques can also provide information. The presence of calcite could indicate the use of fresco.
FTIR has been done and information from the binder has been obtained and calcite has been identified, and it has been considered the use of a fresco. All this information has been included in the manuscript
Rows 490-491: “In this study, none of the samples revealed the presence of any type of binder, perhaps because of low concentrations or the passage of time.” This sentence is not clear. How do the authors establish that the binders are not present? If the pigment is still cohesive onto the stone, it implies that the binder is still there. It is the binder that allows the pigments to be cohesive to the surface. It must be specified. The authors found calcite. Why don't they hypothesize the use of fresco?
FTIR of the samples has been done and the information is included in the manuscript.
The discussions should be improved in order to explain how the results obtained make it possible to date the mural painting before the end of 18th century.
The discussion has been revised in order to include all the comments and suggestions of the reviewers.

Reviewer 2 Report
The research represents an example of a multi analytical and partially non-invasive approach applied to some mural paintings, whose aim is the identification of pigments. The presented research, mainly based on in situ colourimetric and reflectance measurements and laboratory Raman and XRPD investigations, must be improved and needs a significant revision as reported in the specific comments.

Author Response
Thank you for your time and valuable and helpful comments. We have analyzed and revised our manuscript according to your remarks. All changes are indicated in the text. In addition, we send the unified version and version with tracked changes, as the editor requires and our responses below.
The research represents an example of a multi analytical and partially non-invasive approach applied to some mural paintings, whose aim is the identification of pigments. The presented research, mainly based on in situ colourimetric and reflectance measurements and laboratory Raman and XRPD investigations, must be improved and needs a significant revision as reported in the specific comments:
- In order to improve the readability and flow of the sentences, it is recommended to edit the text and the figure captions. The editing process should include both grammar/syntax issues and some typos, such as, for instance, "TiO2" in place of TiO2 end of the paper. Some of the errors are underlined in the following.
The grammar and Figure captions has been revised.
2) Despite the richness in references in the introduction paragraph, results and discussion parts are quite deprived. I would recommend improving the bibliographic references widespread in the aforementioned sections.
Some new references have been added.
- Line 24 - I would not write TiO2, but I would specify the two polymorphs that have been found. In fact, two different crystallographic forms of TiO2 have been revealed in the white and also in the red pigment, i.e. rutile and anatase.
It has been corrected.
Regarding titanium oxides, see all the following comments and revise the abstract, the text and the conclusions accordingly.
The text has been revised according to titanium oxides results.
4) Line 262 I would recommend citing the possible iron oxide-based pigments such as red ochre and hematite (see S. Bruni et al. A non-destructive spectroscopic study of the decoration of archaeological pottery: From matt-painted bichrome ceramic sherds (southern Italy, VIII-VII B.C.) to an intact Etruscan cinerary urn. Spectrochimica Acta Part A Molecular and Biomolecular Spectroscopy (2018)).
It has been revised and the reference included in the manuscript.
5) Line 289 Likewise the previous comment, among the ochre and red Iron-based pigments, I would cite some of the possible different substances such as yellow ochre, goethite, hematite, red ochre, sienna earth, etcetera. The same comments apply to the results of reflectance measurements performed in the laboratory (lines 323-324).
According to the reviewer’s suggestion, the names and chemical formulas of the different Fe-based compounds have been added.
6) Line 295 The authors report that some in situ Raman analyses have been performed, and they refer in brackets to “Figure 4A in situ”, while it would be more precise to refer to Figure “4 in situ”. Actually also, figure “4b in situ” shows some Raman signals that were not cited; morover, those signals deserve to be dicussed since they were evidended in figure “4b in situ”.
It has been changed
7) line 299 - The authors stated that the signal at 1050 cm-1 in spectrum 4a (but it is also present in Figure “4b in situ” is due to “surrounding nitrates”, but, by itself, it colud be also associates to other substances for instance of white lead. The attribution to nitrates requires an explanation.
The sentence has been rewritten to better explain that the band is due to nitrates.
8) Regarding figure “4A in situ”, if captions is correct, ChO should be substituted by ChG. Moreover, figures “4a, 4b, 4c and 4d in laboratory” should be provided with the corresponding color indications, which are reported in the caption.
The colors of the caption refer to the pigment, not to the color of the line of the figure. For clarification, the word “pigment” has been added to the caption in Figure 4.
The pigment ChO is correct in the Figure, then the caption has been changed.
9) Overall, figure 4 is quite confusing. I would suggest editing the picture and referring the 6 Raman spectra to 6 different alphabetical letters. Moreover, I would place both Raman spectra of ChG and CnR obtained in the laboratory just next to the corresponding in situ spectra, interchanging “4b in laboratory” and “4c in laboratory” spectra.
We have updated the figure.
10) Line 361- It is not clear what “considered to be pigments from antiquity” means.
That means that those pigments have been used since antiquity. A new sentence has been added to explain.
11) Line 367- The authors stated that “the formation of azurite )Cu3(CO3)2(OH)2) may result from the transformation of malachite (Cu2CO3(OH)2, but it is known from the literatura that it is rather the opposite.
In effect the formation of malachite is from azurite and not the other way around. It has been changed in the text and a new reference has been added to support the argument.
12) Line 377 - The bands at 609 and 435 cm-1 are more likely to be associated with the presence of rutile rather than hematite (see also comment number 14 down below).
Both signals can be due to iron oxide and also to TiO2. The sentence has been rewritten in order to explain.
13) Line 392 At least one reference for the discussed RR effect and harmonic bands should be given.
A reference has been included.
14) Line 400 The bands at 609 and 444 cm-1 are assigned to rutile correctly, but reference [39] seems misplaced/incorrect here. Please, add the correct reference.
Reference 39 refers to iron oxide identification. A new reference regarding rutile and anatase has been added.
15) Line 403 Paragraph 4.2.3 is poorly describing the plenty of information coming from diffractograms of figure 5; regarding the ChR sample, at least peaks of anatase and rutile should be evidenced, especially because they confirmed what the reviewer suggested about previously discussed Raman results of ChR sample.
The paragraph has been rewritten adding more information.
16) Line 419 X-ray diffraction (XRD) is incorrect since paragraph 4.2.4 is about SEM-EDX analyses
Thank you for the correction. It has been corrected.
17) Line 462 The discussion about the red sample should be revised (see also comment 20 down below). In fact, the red sample contains both rutile and anatase and, as a consequence, they have to be mentioned and discussed. Moreover, in line 466 authors state that TiO2 may be due to "previous painting", which seems quite weird.
We agree with the comment and this part has been revised and rewritten.
18) Line 483-491 All the discussion about binders, although correct, seems to be quite pleonastic here, since, in the end, none of the utilised analytical techniques might easily detect any of the cited organic binders; the authors conclude with “ none of the samples revealed the presence of any type of binder, perhaps because of low concentrations or thepassage of time”, but it is likely that this lacking of results may also have occurred because no FTIR or GC-MS analyses were carried out.
FTIR has been included and the discussion has been revised.
19) Line 501 The presence of calcite might also be due to the fresco technique that the authors mention in the paper.
After the FTIR information, the binder was identified as calcite and gypsum and this is included in the manuscript.
20) Line 511 The discussion about sample ChW should be revised since the authors declare that “titanium oxide (TiO2) in the rutile phase as the fundamental component of the White pigment” while the time reference at line 517 actually refers to anatase. In the revising process, the authors should also take into account that rutile, differently from anatase (which is quite rare as a mineral), also occurs in nature.
Thank you for the comment. It has been revised the manuscript
21) Line 540 - The statement "The ochre pigment originated from the earth, whereas the red pigment is a calcined ochre" seems quite audacious. There is massive literature about the natural or synthetic origin of red ochre/hematite pigments, and none is mentioned here for supporting that conclusion.
This paragraph has been rewritten.

Reviewer 3 Report
Dear Authors,
The case study you present of the analytical investigation of a mural from the Cathedral of Santo Domingo is interesting for readers given its historical importance. The methodology of in-situ and lab measurements are not innovative nor are the results. To strengthen this paper a summary table could be added so that it is directly understood what was found with each colour in-situ and in the lab with the analytical instrumentations. Specifically, it could be better to discuss the results by sample rather than technique.
Author Response
Thank you for your time and valuable and helpful comments. We have analyzed and revised our manuscript according to your remarks. All changes are indicated in the text. In addition, we send the unified version and version with tracked changes, as the editor requires and our responses below.
The case study you present of the analytical investigation of a mural from the Cathedral of Santo Domingo is interesting for readers given its historical importance.
New information has been included in the Introduction.
The methodology of in-situ and lab measurements are not innovative nor are the results. To strengthen this paper a summary table could be added so that it is directly understood what was found with each colour in-situ and in the lab with the analytical instrumentations.
A Table with the information is included in the manuscript.
Specifically, it could be better to discuss the results by sample rather than technique.
We agree with the reviewer, therefore, in the discussion the results are analyzed by samples, although they have been discussed by techniques in the experimental and results parts.

Round 2
Reviewer 2 Report
The manuscript still needs some minor revisions, as reported in the following:
1) Figure 4 was corrected. Nevertheless, the reviewer still does not understand why in figure 4b (to be the clearest possible, in the middle of figure 4b) there is a label reporting “ChO”. If the spectrum reported in figure 4b is the spectrum of the green sample obtained in situ, the label in the middle of figure 4b should be corrected with ChG in place of ChO, as previously required.
Otherwise, the authors should courteously explain why placing that ChO label in a box that reports the spectrum of the green sample.
2) Line 439-446 – The FTIR bands at about 3000 cm-1, clearly visible in figure 5, should be discussed; some conclusions about their presence should also be provided.
Author Response
The manuscript still needs some minor revisions, as reported in the following:
1) Figure 4 was corrected. Nevertheless, the reviewer still does not understand why in figure 4b (to be the clearest possible, in the middle of figure 4b) there is a label reporting “ChO”. If the spectrum reported in figure 4b is the spectrum of the green sample obtained in situ, the label in the middle of figure 4b should be corrected with ChG in place of ChO, as previously required.
Thank you for the comment. The Figure has been corrected, since the Figure 4b corresponded to the ChO sample.
Otherwise, the authors should courteously explain why placing that ChO label in a box that reports the spectrum of the green sample.
We apologise, because the Figure 4b corresponds to the ChO sample and was not corrected in the previous revision.
2) Line 439-446 – The FTIR bands at about 3000 cm-1, clearly visible in figure 5, should be discussed; some conclusions about their presence should also be provided.
This bands have been explained in the new version of the text.
Reviewer 3 Report
Dear authors,
Main changes indicated by the reviewers have been carried out adequately.
I suggest only to shift the position of table 1 to the start of section 4 so that a summary of results is immediately available for reference prior to diuscussions..
Author Response
Dear authors,
Main changes indicated by the reviewers have been carried out adequately.
I suggest only to shift the position of table 1 to the start of section 4 so that a summary of results is immediately available for reference prior to diuscussions.
The Table has been moved to the position suggested by the reviewer.